# Timing of Bisphosphonate (Alendronate) Initiation after Surgery for Fragility Fracture: A Population-Based Cohort Study

**DOI:** 10.3390/jcm10122541

**Published:** 2021-06-08

**Authors:** Meng-Huang Wu, Yu-Sheng Lin, Christopher Wu, Ching-Yu Lee, Yi-Chia Chen, Tsung-Jen Huang, Jur-Shan Cheng

**Affiliations:** 1Department of Orthopedics, Taipei Medical University Hospital, Taipei 110301, Taiwan; maxwutmu@gmail.com (M.-H.W.); ejaca22@gmail.com (C.-Y.L.); tjdhuang@tmu.edu.tw (T.-J.H.); 2Department of Orthopaedics, School of Medicine, College of Medicine, Taipei Medical University, Taipei 110301, Taiwan; 3Department of Cardiology, Chang Gung Memorial Hospital, Chiayi 613016, Taiwan; dissert@cgmh.org.tw; 4College of Medicine, Chang Gung University, Taoyuan 333323, Taiwan; 5College of Medicine, Taipei Medical University, Taipei 110301, Taiwan; cjwuchris@gmail.com; 6Research Services Center for Health Information, Chang Gung University, Taoyuan 333323, Taiwan; chia0317@mail.cgu.edu.tw; 7Clinical Informatics and Medical Statistics Research Center, College of Medicine, Chang Gung University, Taoyuan 333323, Taiwan; 8Department of Biomedical Sciences, College of Medicine, Chang Gung University, Taoyuan 333323, Taiwan; 9Department of Emergency Medicine, Chang Gung Memorial Hospital, Keelung 204201, Taiwan

**Keywords:** fragility fracture, initiation timing, second fracture, adverse events

## Abstract

Bisphosphonates are used as first-line treatment for the prevention of fragility fracture (FF); they act by inhibiting osteoclast-mediated bone resorption. The timing of their administration after FF surgery is controversial; thus, we compared the incidence of second FF, surgery for second FF, and adverse events associated with early initiation of bisphosphonates (EIBP, within 3 months of FF surgery) and late initiation of bisphosphonates (LIBP, 3 months after FF surgery) in bisphosphonate-naïve patients. This retrospective population-based cohort study used data from Taiwan’s Health and Welfare Data Science Center (2004–2012). A total of 298,377 patients received surgeries for FF between 2006 and 2010; of them, 1209 (937 EIBP and 272 LIBP) received first-time bisphosphonates (oral alendronate, 70 mg, once a week). The incidence of second FF (subdistribution hazard ratio (SHR) = 0.509; 95% confidence interval (CI): 0.352–0.735), second FF surgery (SHR = 0.452; 95% CI: 0.268–0.763), and adverse events (SHR = 0.728; 95% CI: 0.594–0.893) was significantly lower in the EIBP group than in the LIBP group. Our findings indicate that bisphosphonates should be initiated within 3 months after surgery for FF.

## 1. Introduction

Bisphosphonates are commonly used as first-line treatment for the prevention of primary and second fragility fractures (FFs). Bisphosphonates prevent FFs by inhibiting osteoclast-mediated bone resorption, thus increasing bone strength and preventing bone loss. They are endocytosed by osteoclasts and inhibit the activity of farnesyl pyrophosphate synthase, which is responsible for the production of cholesterol and intracellular localization of GTPase signaling proteins, which are essential for osteoclast functions [1]. However, osteoclasts play a vital role in callus formation and bone remodeling as well as remodeling of the callus into cortical bone [2,3]. Thus, bisphosphonates may exert adverse effects on fracture healing. However, animal studies have revealed dissimilar results: no effect on fracture healing [4,5], delayed fracture healing [6,7,8], and enhanced fracture healing [9,10,11,12,13]. In a case–control study, Li et al. stated that bisphosphonate use in the postfracture period was linked to increased likelihood of nonunion [14]. However, recent research in humans has indicated that early bisphosphonate administration does not affect the rate of healing of fractures, clinical outcomes, or the complications rate [15,16]. No whole-population-based study has examined this question.

We hypothesized that early bisphosphonate initiation has better outcomes than late bisphosphonate initiation. Therefore, we compared the incidence of second FF, surgery for second FF, and adverse events associated with early initiation of bisphosphonates (EIBP; within 3 months of FF surgery) and late initiation of bisphosphonates (LIBP; 3 months after FF surgery) used as an oral alendronate (70 mg once a week).

## 2. Materials and Methods

### 2.1. Data Sources and Study Samples

This retrospective cohort study used national data (2004–2012) including National Health Insurance (NHI) claims data, the Cancer Registry Database, and the Death Registry Database. More than 99% of the population is enrolled in the mandatory, single-payer NHI program, which provides comprehensive care, including outpatient and inpatient services, laboratory tests, and prescription drugs.

We include patients aged ≥50 years who underwent surgery for FF (spine, hip, upper limb, or lower limb fracture) during 2006–2010 (the index surgery and date) and received bisphosphonate monotherapy with medication possession ratio (MPR) ≥ 0.5 within the 1 year after the surgery (Figure 1, Table 1). We excluded patients with pathologic fracture, cancer history, traffic accident injury, multiple fractures, or history of taking antiosteoporotic medicine. In addition, patients with respiratory failure, pneumonia, an acute cardiac event, a cerebrovascular accident and spinal cord injury, nerve injury, renal failure, upper gastrointestinal (UGI) bleeding, acute postoperative hemorrhage, urinary tract infection, sepsis, osteomyelitis, postoperative infection, or nonunion or malunion within the 3 months before the index date were excluded. The index date was defined as the day of the first FF surgery. The selected participants were divided into EIBP and LIBP groups. To improve the comparability of the groups, the EIPB group was 1:1 matched with the LIBP group through propensity score matching by using a logistic model to estimate the probability of initiating the therapy more than 3 months after the index date. The covariates included in the model were age (<65 and ≥65 years), sex (male and female), Charlson comorbidity index (CCI; 0, 1, and ≥2), and FF surgery site (spine, hip, upper limb, and lower limb). All patients had a minimum 2-year follow-up.

### 2.2. Confounders and Bias

Registration and classification bias of fractures may be present in the NHI claims data of patients receiving FF surgery; patients may also have higher frequency of complications and reoperation following FFs, thus artificially inflating the fracture rate. To limit the possible biases, we introduced a 90-day “washout” period for each fracture and adverse event; thus, any new fracture with the same ICD-9 code or adverse event associated with surgery was ignored. However, the NHI claims data on fractures do not specify the side of appendicular fractures (i.e., left or right). Furthermore, fractures to the axial skeleton are classified only as lumbar, thoracic, or cervical. This may result in the exclusion of fractures to, for example, the left tibia, if a right tibial fracture was registered in the preceding 90 days. We risked underestimating the fracture rate. We could not correct for this possible bias.

Medications, including thyroxine, proton pump inhibitors, and steroids, affect the likelihood of FF and adverse events. The prescription of these medications for more than 3 months was recorded.

### 2.3. Statistical Analysis

All statistical analyses were conducted using SAS (version 9.4; SAS Institute, Cary, NC, USA). The incidence of outcomes in the EIBP and LIBP groups was estimated by calculating the number of cases per 100 person-years. Person-years were calculated using the time from the index date to the date of the event or the end of follow-up (31 December 2012), whichever occurred first. The cumulative incidence of the outcomes in the EIBP and LIBP groups was estimated using the modified Kaplan–Meier method and the Gray method, which considers death a competing risk event [17].

Subdistribution hazard models that accounted for competing mortality were used to estimate the adjusted subdistribution hazard ratio (SHR) of outcomes, with adjustments for age (<65 and ≥65 years old); sex (male and female); site of the index FF surgery (spine, upper limb, lower limb, and hip); CCI (0, 1, and ≥2); and comorbidities (cerebrovascular disease, chronic pulmonary disease, rheumatologic disease, peptic ulcer disease, diabetes, and renal disease). Risks of secondary FF, second FF surgery, and revision surgery in the EIBP and LIBP groups were also compared among the subgroups of the index FF surgery site and age [18].

If significant differences were noted in initial FF surgery and age between the groups (Table 1), we performed a subgroup analysis to evaluate the effect of initial FF surgery and age on the outcomes. Although the incidence of rheumatologic disease differed between the groups, we did not evaluate the effect of this disease because of the small number of patients.

This study was granted ethical approval by the Institutional Review Board of Chang Gung Memorial Hospital of Taiwan (104-3677B).

## 3. Results

### 3.1. Baseline Characteristics

A total of 130,406 patients underwent FF surgery during 2006–2010. Among them, 12,863 received antiosteoporotic medicine. We identified 1209 patients who were first-time users of bisphosphonate (oral alendronate, 70 mg once a week) after FF surgery and who had an MPR of ≥0.5. Of these patients, 937 were included in the EIBP group and 272 in the LIBP group. After propensity score matching, both groups contained 262 patients (Table 1, Figure 1); these groups had similar sex distribution, CCIs, and medical history. More patients in the EIBP group had cerebrovascular disease than in the LIBP group (4.20% vs. 1.15%, *p* = 0.030).

Significant differences were observed in the incidence of second FF, second FF surgery, and adverse events between the EIBP and LIBP groups (Table 2). A higher incidence of second FF (32.44% vs. 19.08%, *p* = 0.0005) and second FF surgery (18.70% vs. 9.16%, *p* = 0.0021) was observed in the LIBP group.

A higher incidence of second FF surgery was detected in the LIBP group in patients with complications of respiratory failure (13.7% vs. 5.344%, *p* = 0.0025), renal failure (6.87% vs. 2.29%, *p* = 0.0174), and UGI bleeding (75.95% vs. 26.34%, *p* = 0.0361).

### 3.2. Second FF and Second FF Surgery

Multivariate subdistribution hazard regression analyses of factors associated with second FF, revision surgery, and second FF surgery were conducted (Figure 2). The findings indicated that the incidences of second FF (SHR = 0.509; 95% confidence interval (95% CI): 0.352–0.735, *p* = 0.0003) and second FF surgery (SHR = 0.452; 95% CI: 0.268–0.763, *p* = 0.0029) were significantly lower in the EIBP group than in the LIBP group (Figure 2 and Figure 3). Patients aged <65 years (SHR = 1.632; 95% CI: 1.044–2.549, *p* = 0.0315), who had cerebrovascular disease (SHR = 3.182; 95% CI: 1.174–8.626, *p* = 0.0229), and who had renal disease (SHR = 5.259; 95% CI: 1.370–20.186, *p* = 0.0156) had significantly higher incidence of second FF. No significant difference was observed in any factor for incidence of revision surgery between the EIBP and LIBP groups (Figure 4).

### 3.3. Adverse Events after FFs

After initial FF surgery, the incidence of adverse events (Figure 4)—respiratory failure (SHR = 0.363; 95% CI: 0.189–0.697, *p* = 0.0024), renal failure (SHR = 0.346; 95% CI: 0.134–0.894, *p* = 0.0284), UGI bleeding (SHR = 0.76; 95% CI: 0.618–0.935, *p* = 0.0095), and overall complications (SHR = 0.7280; 95% CI: 0.5940–0.8930, *p* = 0.0023)—was lower in the EIBP group than in the LIBP group. The incidence of respiratory failure (SHR = 0.316; 95% CI: 0.174–0.572, *p* = 0.0001) and renal failure (SHR = 0.354; 95% CI: 0.140–0.895, *p* = 0.0282) was higher in women, but that of UGI bleeding (SHR = 1.508; 95% CI: 1.137–2.000, *p* = 0.004) and overall complications (SHR = 1.380; 95% CI: 1.051–1.810, *p* = 0.020) was higher in men.

A lower incidence of respiratory failure was noted in patients with upper limb FF surgery (SHR = 0.421; 95% CI: 0.196–0.904, *p* = 0.0024) and UGI bleeding (SHR = 0.79; 95% CI: 0.625–0.998, *p* = 0.0479), and a higher incidence in patients aged > 65 years (SHR = 7.281; 95% CI: 2.029–26.12, *p* = 0.0023). Peptic ulcer was associated with a higher incidence of complications (SHR = 1.85; 95% CI: 1.131–3.023, *p* = 0.0143).

## 4. Discussion

Osteoporosis is a common skeletal disorder and is associated with osteoporotic fractures, which lead to a considerable rise in the likelihood of morbidity and mortality as well as increased health care costs. Osteoporosis treatment can increase bone strength and decrease fracture risk. One of the drugs most commonly used to treat osteoporosis is bisphosphonate [19], especially alendronate due to its cost effectiveness and support from well-established evidence. Denosumab was popular, but it can cause rebound osteoporosis and increased risk of spine fracture after discontinuation, which has led to a decrease in use [20]. Therefore, recent studies have proposed a re-evaluation of bisphosphonate use. The most appropriate timing of bisphosphonate initiation remains debatable. Our data indicated that early initiation of oral alendronate was associated with lower incidence of second FF, second FF surgery, and adverse events than late initiation.

Several randomized clinical trials have investigated the effects of the early postoperative use of bisphosphonates (<3 months) on FFs. Eriksen et al. suggested bisphosphonate initiation (zoledronic acid) 2–12 weeks after surgery to decrease the incidence of clinical vertebral fracture, nonvertebral hip fracture, and mortality [21]. Lyles et al. found similar results when zoledronic acid was initiated within 90 days of hip fracture surgery [22]. Uchiyama et al. stated that early administration of alendronate did not delay fracture healing and decreased bone turnover [23]. A meta-analysis conducted by Li et al. included 253 individuals from four randomized clinical trials (RCTs) with measurements of healing time and 2365 participants from six RCTs with measurements of delayed or nonunion rates of fracture healing. The meta-analysis thus included 10 studies with 2888 patients. Four of the trials used alendronate, three used zoledronic, two used risedronate, and one used etidronate. No significant difference was observed in radiological fracture healing time between the bisphosphonate therapy group and control group (mean difference = 0.47, 95% CI: −2.75 to 3.69). In previous studies, early bisphosphonate administration resulted in higher bone mineral density (BMD). HORIZON recurrent fracture trials revealed that patients receiving bisphosphonate 6 weeks to 3 months after hip fracture had higher total hip and femoral neck BMD at 12 months than those receiving a bisphosphonate before 6 weeks [21]. This may be because bisphosphonates do not directly affect cells involved in the inflammatory phase (i.e., osteoblasts) or the formation of soft and hard callus; instead, bisphosphonates interrupt the remodeling of hard callus. Consequently, bisphosphonate initiation within 3 months after surgical treatment of FFs may be more effective timing for preventing second FFs and second FF surgeries.

Our cohort had lower incidence of respiratory failure, UGI bleeding, renal failure, and other complications. This may have been due to the faster bone integration or stabilization of implants, allowing early ambulation and preventing respiratory failure due to aspiration pneumonia.

Alendronate use has been demonstrated to decrease the risk of cardiovascular events. Sing et al. observed significantly lower risk of 1-year cardiovascular mortality (SHR 0.33; 95% CI: 0.17–0.65) and incident myocardial infarction (SHR 0.55; 95% CI: 0.34–0.89) following hip fracture [24]. This is due to nitrogen-containing bisphosphonates targeting the mevalonate pathway, specifically farnesyl pyrophosphate synthase, which has the same pathway as statins [25]. Alishiri et al. concluded that in patients with aortic stenosis and concurrent osteoporosis, alendronate treatment retarded stenosis progression and improved the outcome [26].

The lower risk of UGI bleeding may be related to the indication of bisphosphonate. Therefore, physicians should be cautious about using ulcer-inducing agents in patients with EIBP. Alendronate causes oxidative gastric damage by increasing myeloperoxidase activity and lipid peroxidation, which cause the formation of gastric ulcers and impair gastric ulcer healing [27]. In a population-based study, Peng et al. concluded that alendronate was a risk factor for UGI bleeding (SHR = 1.32; 95% CI: 1.02–1.71) [28]. This was inconsistent with our results, likely because EIBP may exacerbate certain symptoms in patients and is therefore usually avoided.

Bisphosphonate is contraindicated in patients with renal insufficiency because of the risk of renal function deterioration [29]. In our cohort, the risk of renal failure was lower in the EIBP group likely due to the avoidance of other medications causing renal toxicity due to shorter hospital stay. Moreover, improvement of function leads to lower nonsteroidal anti-inflammatory drug use, which may be another explanation for the finding.

Patients with EIBP have earlier physician attention and therefore have more awareness of their general condition because of programs such as the fracture liaison service (FLS). The FLS has improved the quality of care in patients with fracture and helped prevent secondary fractures [30]. This may be another reason for the fewer complications in the EIBP group. In a nationwide cohort study, Wang et al. reported that early initiation (15–84 days) of antiosteoporosis medicine reduced the risk of fracture-related hospitalization [31]. These findings also indicate the importance of the FLS in identifying patients suitable for ELBP.

The strengths of this study include the completeness of patient follow-up and high accuracy of coding in the national health database [32]. The study had some limitations. First, the NHI claims database lacks information on the Fracture Risk Assessment Tool score, BMD, smoking, compliance, vitamin D deficiency, and bone turnover markers. Second, we included only patients who received first-time alendronate. Third, we could not differentiate the causes of fractures, which may also have included atypical fractures. The incidence of atypical femoral fractures was found to be between 0.3 and 85.9 per 100,000 person-years [33,34]. Short- or long-term alendronate use is not significantly associated with higher risk of atypical femoral fractures [35]. Therefore, the low frequency of atypical fractures probably did not affect the study outcomes. Finally, we could only evaluate the associations of early and late bisphosphonate use with adverse events and could not follow up on the intergroup difference in perioperative complications that would have been attributable to bisphosphonate initiation. Prospective studies should be conducted to analyze this.

## 5. Conclusions

Bisphosphonate initiation within 3 months of FF surgery is associated with lower incidence of second FF, second fracture surgery, and adverse events and should therefore be considered over late initiation in these patients.

## Figures and Tables

**Figure 1 jcm-10-02541-f001:**
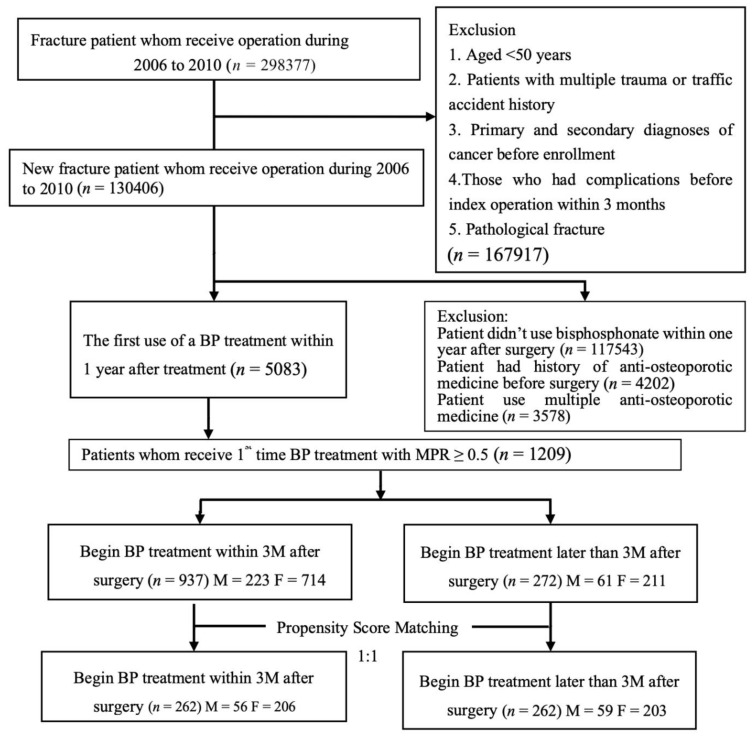
Study protocol. BP: bisphosphonate (alendronate); MPR: medication possession ratio.

**Figure 2 jcm-10-02541-f002:**
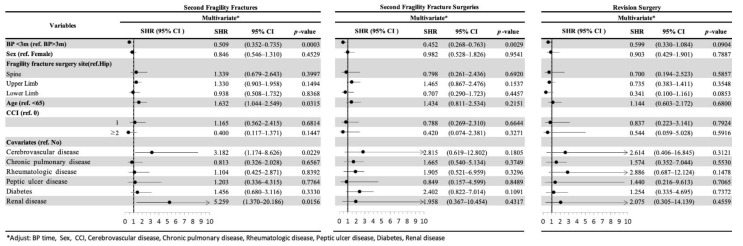
Risk factors and adjusted hazard ratio of fragility fractures stratified by bisphosphonate initiation time, sex, age, Charlson comorbidity index, and comorbidities, by using a proportional subdistribution hazards model. BP: bisphosphonate initiation; CCI: Charlson comorbidity index; CI: confidence interval.

**Figure 3 jcm-10-02541-f003:**
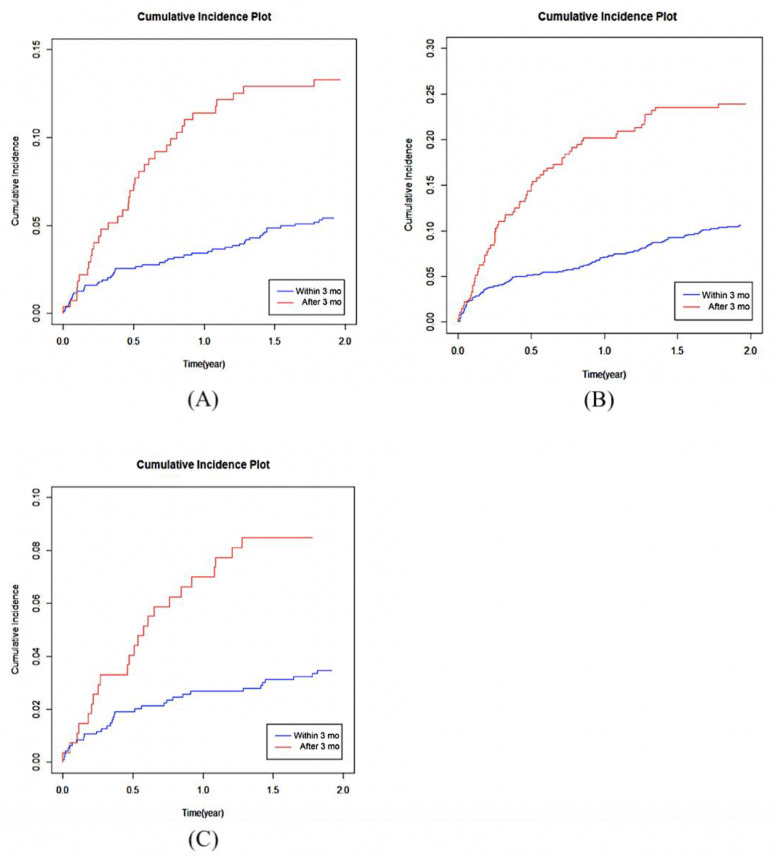
Cumulative incidence of (**A**) second fragility fracture, (**B**) second fragility fracture surgery, and (**C**) revision surgery in patients with bisphosphonate initiation within 3 months after fragility fracture surgery versus more than 3 months after fragility fracture surgery.

**Figure 4 jcm-10-02541-f004:**
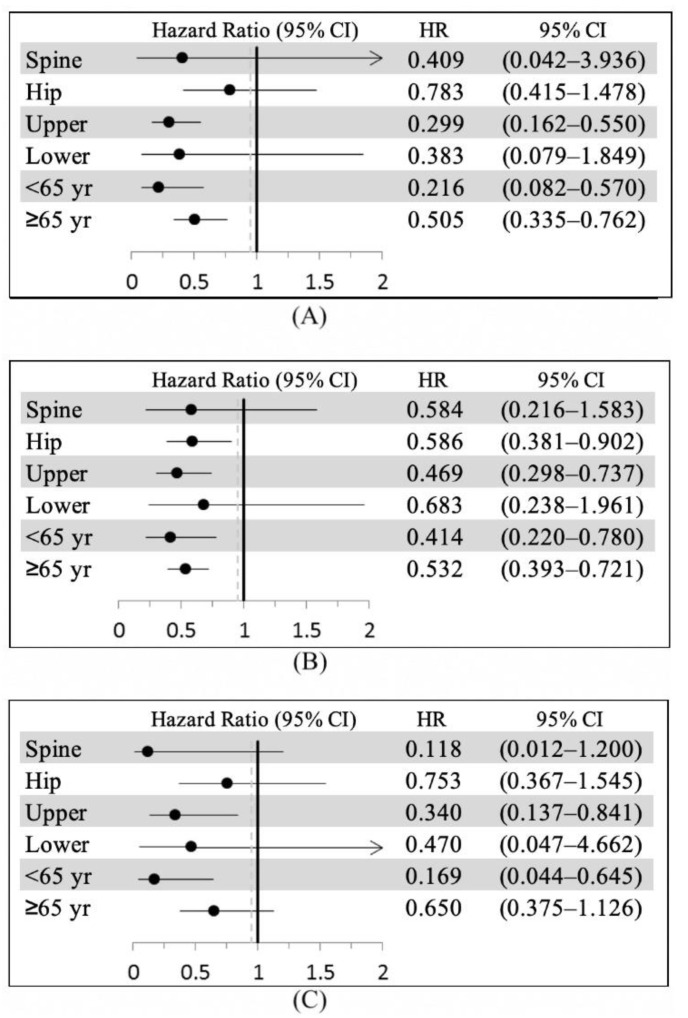
Forest plot of (**A**) second fragility fracture, (**B**) second fragility fracture surgery, and (**C**) revision surgery in patients with bisphosphonate initiation within 3 months after fragility fracture surgery versus more than 3 months after fragility fracture surgery for different fragility fracture types and age groups. SHR, subdistribution hazard ratio; CI, confidence interval; Upper, upper limb; Lower, lower limb.

**Table 1 jcm-10-02541-t001:** Clinicodemographic characteristics of the study population.

Variables		Total		EIBP		LIBP		*p* Value
		(*n* = 524)		(*n* = 262)		(*n* = 262)		
		Count	%	Count	%	Count	%	
Sex								
	Male	115	21.946	56	21.374	59	22.519	0.752
	Female	409	78.053	206	78.626	203	77.481	
Age (years)								
	<65	129	24.618	65	24.809	64	24.427	0.919
	≥65	395	75.382	197	75.191	198	75.573	
CCI								
	0	327	62.405	163	62.214	164	62.595	0.968
	1	123	23.473	61	23.282	62	23.664	
	≥2	74	14.122	38	14.504	36	13.740	
Covariates								
	Cerebrovascular disease	14	2.672	11	4.198	3	1.145	0.030
	Chronic pulmonary disease	43	8.206	21	8.015	22	8.397	0.874
	Rheumatologic disease	21	4.008	8	3.053	13	4.962	0.265
	Peptic ulcer disease	20	3.817	10	3.817	10	3.817	1.000
	Diabetes	104	19.847	51	19.466	53	20.229	0.827
	Diabetes with chronic complications	26	4.962	13	4.962	13	4.962	1.000
	Renal disease	13	2.481	6	2.290	7	2.672	0.779
	Obesity	5	0.954	3	1.145	2	0.763	0.744
	History of fragility fracture	42	8.015	22	8.396	20	7.633	0.615
Fragility fracture surgery site								
	Spine	38	7.252	19	7.252	19	7.252	1.000
	Hip	224	42.748	112	42.748	112	42.748	
	Upper limb	188	35.878	94	35.878	94	35.878	
	Lower limb	74	14.122	37	14.122	37	14.122	
Concomitant drug								
	Proton pump inhibitor	4	0.331	3	0.320	1	0.368	0.905
	Steroid	30	2.481	19	2.028	11	4.044	0.059
	Antiepileptic	4	0.331	3	0.320	1	0.368	0.905
Follow-up time (mean years)		4.314		4.2456		4.3824		

CCI, Charlson comorbidity index; EIBP, early initiation of bisphosphonates (within 3 months of surgery); LIBP, late initiation of bisphosphonates (3 months after surgery).

**Table 2 jcm-10-02541-t002:** Incidence of second fragility fractures and surgeries, revision surgery, death, and complications.

Variables		EIBP			LIBP			Log-Rank *p* Value	Chi-Square *p* Value
		(*n* = 262)			(*n* = 262)				
		Count	%	Incidence *	Count	%	Incidence *		
Site of second fragility fracture		50	19.084	5.173	85	32.443	9.693	0.0005	0.0005
	Trunk	<5	<5.000		<5	<5.000			
	Spine	8	16.000		22	25.882			
	Hip	22	44.000		31	36.471			
	Upper Limb	16	32.000		23	27.059			
	Lower Limb	<5	<5.000		<5	<5.000			
Site of second fracture surgery		24	9.160	2.323	49	18.702	4.943	0.0021	0.0016
	Spine	<5	<5		5	10.204			
	Hip	19	79.167		24	48.980			
	Upper Limb	4	16.667		14	28.571			
	Lower Limb	<5	<5		5	10.204			
Site of revision surgery		18	6.870	1.705	30	11.450	2.858	0.0789	0.0692
	Spine	<5	<5.000		<5	<10.000			
	Hip	15	83.333		11	36.667			
	Upper Limb	<5	<15.000		14	46.667			
	Lower Limb	<5	<5.000		<5	<5.000			
Death		35	13.359	3.146	51	19.466	4.442	0.1351	0.0591
Complication									
	Respiratory failure	14	5.344	1.271	36	13.740	3.208	0.0025	0.0011
	Pneumonia	81	30.916	8.577	93	35.496	9.974	0.3032	0.2657
	Acute cardiac event	55	20.992	5.554	65	24.809	6.631	0.3111	0.2985
	CVA and SCI, nerve injury	69	26.336	7.211	67	25.573	6.948	0.8965	0.842
	Renal failure	6	2.290	0.543	18	6.870	1.597	0.0174	0.0122
	UGI bleeding	170	64.885	11.738	199	75.954	35.809	0.0361	0.0055
	Acute postoperative hemorrhage	11	4.198	0.989	12	4.580	1.045	0.8927	0.8311
	Urinary tract infection	97	37.023	11.210	115	43.893	13.123	0.2457	0.1091
	Sepsis	32	12.214	2.989	38	14.504	3.445	0.6182	0.441
	Osteomyelitis	5	1.908	0.454	13	4.962	1.170	0.0604	0.055
	Postoperative Infection	5	1.908	0.457	13	4.962	1.182	0.059	0.055
	Nonunion/malunion	6	2.290	0.549	12	4.580	1.094	0.1581	0.1501

* per 100 person-years. EIBP, early initiation of bisphosphonates (within 3 months of surgery); LIBP, late initiation of bisphosphonates (3 months after surgery); CCI, Charlson comorbidity index; CVA, cerebrovascular accident; SCI, spinal cord injury; UGI, upper gastrointestinal.

## Data Availability

The data presented in this study are available on request from the corresponding author.

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
