# Peer review of "Timing of Bisphosphonate (Alendronate) Initiation after Surgery for Fragility Fracture: A Population-Based Cohort Study"

_jcm, 2021, doi:10.3390/jcm10122541_

Round 1

Reviewer 1 Report

The manuscript entitled “Timing of Bisphosphonate (Alendronate) Initiation after Surgical Treatment of Fragility Fractures—A Population-Based Cohort Study” aimed to assess how the timing of alendronate treatment (within 3 months-early initiation/after 3 months after fracture surgery-late initiation) affects patients' outcomes. The study found that the incidence of second fragility fractures and second fracture surgeries was higher in late initiation group. The incidence of adverse events was lower in early initiation group.

The manuscript is generally well written, aims and methods are clearly stated. Methods and statistical analysis are sound and the results are presented and described well. However, I have some minor comments and suggestions.

Introduction could be improved. It could focus on evidence published specifically on alendronate so far. For example the occurrence of cardiac events after alendronate treatment could be mentioned as well. In similar direction, Discussion could contain more studies, which focused on alendronate.

Next, could authors explain why they did not include acute cardiac event as one of the adverse events to be followed up?  

If the mortality was taken into account as a competing event in the incidence of fracture, why it was not taken into account in the regression analysis aswell? Then, not the Cox regression analysis should be used, however the competing-risks regression based on Fine and Gray’s proportional subhazards model and SHR should be reported.

Technical comments:

  • Table 1. – a typo in % for gender in total population (176.527%). Please check and correct it.
  • Please add the mean age with standard deviations (for total sample, EIBP, LIBP).
  • Table 1. Could you add a variable on a “Follow-up time” (Median, IQR ranges) (for total sample, EIBP, LIBP)?
  • Table 1. general comment- How did you handle missing values?
  • Table 2.-The listing of the complications in Table 2 could me misleading to the reader, because it seems that patients were followed up for these complications. However, patients with these complications were excluded before the index date.
  • Could you clarify the index date in the manuscript?
  • Information provided in Chapter 2.2 and 2.4 (Statistical analysis) overlap. Please concise it and provide only one Chapter for the Statistical analysis following guidelines for the J Clin Med.

Reviewer 2 Report

Thank you for the opportunity to provide a review of your manuscript, which addresses a clinical question of significant relevance.  Your findings   On this large National database review suggest that early pharmacologic treatment with bisphosphonates verses later treatment improves the outcomes and reduces risk of secondary fragility fractures.

The introduction would benefit from a hypothesis. 

The covariates assessed did not include potential variables of relevance, including prior history of fragility fracture, steroid use, smoking, use of antiepileptic medication, or obesity.  

What was the rationale for excluding patients with UTI or peptic ulcer and the other exclusions? I'm not certain I understand why these were chosen as exclusion criteria. 

If the cohorts were divided by whether BP's were started within 3 months of the FF, it does not follow that there would be a difference in perioperative complications between the groups that would be attributable to BP initiation. 

Is there any way in your database to assess patient compliance with medication? All that I can glean from the study is whether medication was prescribed within 3 months or not. 

Vitamin D deficiency was not assessed and could impact the initial and secondary fracture risk. Is data available for this? 

Another aspect of fragility fracture management is appropriate screening (DEXA). Is this information available? 

How were fragility fractures identified? The exclusion criteria are noted, but how was it determined that the remaining fractures occurred by low-energy mechanisms? 

Based on the methods, it seems that a minimum 2-yr follow-up was performed. Is this correct? If so it would be helpful to be explicit about that. 

Round 2

Reviewer 1 Report

I thank the authors for taking effort in responding my comments. In large part, my questions have been addressed. However, I still think that there are some uncertainties regarding the adverse events for which the patients have been followed up. In Lines 70-74 authors state, that patients suffering particular events were excluded before index date and they refer to Table 2. There are the complications (=adverse events) listed. However, it is not clear, for which particular adverse events were then the patients followed up. Was it the same list of events as for which they were excluded before index date? It makes it difficult to judge also due to the fact that the Results for the adverse events in Lines 234-247 are not supported by the Table nor by the graphical illustration (Forest plot). Could the authors clarify this?

With regard to my previous comment on competing risk regression, if the authors used sub-distribution hazards models then I recommend authors to report the SHR throughout the manuscript instead of HR.

I would like to ask the authors to add the units reported (days or months) to the follow up times (Table 1). Lastly, if authors do not have data on mean age and SD of the sample, then they should not include it in the Table 1. Now it reads as with empty cell for Mean age and SD. Please delete that row.

Reviewer 2 Report

Thank you for the opportunity to review your revised admission.  It appears that the comments have been adequately addressed, though some limitations remain.  The majority of these have reasoning given as to why the limitations could not be addressed in this study.  There are still residual language editing changes that should be made before this could be accepted fully for publication.

Author Response

Thank you for the opportunity to review your revised admission.  It appears that the comments have been adequately addressed, though some limitations remain.  The majority of these have reasoning given as to why the limitations could not be addressed in this study.  There are still residual language editing changes that should be made before this could be accepted fully for publication

Answer: Thank you for your comments. We have sent our manuscript for language editing as you suggested.